# DIVERSITY-DRIVEN EXPLORATION STRATEGY FOR DEEP REINFORCEMENT LEARNING

**Zhang-Wei Hong, Tzu-Yun Shann, Shih-Yang Su, Yi-Hsiang Chang, and Chun-Yi Lee**
*williamd4112@gapp.nthu.edu.tw, ariel@shann.net, at7788546@gmail.com, s106062520@m106.nthu.edu.tw, cylee@cs.nthu.edu.tw*
Department of Computer Science, National Tsing Hua University, Hsinchu, Taiwan

## ABSTRACT

Efficient exploration remains a challenging research problem in reinforcement learning, especially when an environment contains large state spaces, deceptive local optima, or sparse rewards. To tackle this problem, we present a diversity-driven approach for exploration, which can be easily combined with both off- and on-policy reinforcement learning algorithms. We show that by simply adding a distance measure to the loss function, the proposed methodology significantly enhances an agent's exploratory behaviors, and thus preventing the policy from being trapped in local optima. We further propose an adaptive scaling method for stabilizing the learning process. Our experimental results in Atari 2600 show that our method outperforms baseline approaches in several tasks in terms of mean scores and exploration efficiency.

## 1 INTRODUCTION

In deep reinforcement learning (DRL), a common approach for exploration is to adopt simple heuristic methods, such as $\epsilon$-greedy Sutton & Barto (1998); Mnih (2015) or entropy regularization Mnih (2016). However, such strategies are unlikely to yield satisfactory results in tasks with deceptive or sparse rewards Fortunato (2018); Osband et al. (2017). A more sophisticated line of methods provides agents with bonus rewards whenever they visit a novel state Stadie et al. (2015); Pathak et al. (2017). While being effective, these algorithms often require statistical or predictive models to evaluate the novelty of a state, and therefore increase the complexity of the training procedure. Some researchers borrow the idea of random perturbation from evolutionary algorithms Plappert (2018); Fortunato (2018). Despite its simplicity, these methods are less efficient in large state spaces, since random noise changes the behavioral patterns of agents in an unpredictable fashion Fortunato (2018); Plappert (2018). We present a diversity-driven exploration strategy, a methodology that encourages a DRL agent to attempt policies different from its prior policies. We propose to use a distance measure to modify the loss function to tackle the problems of large state spaces, deceptive local optima, and sparsity in reward signals. The distance measure evaluates the novelty between the current policy and a set of prior policies. We demonstrate that our methodology is complementary and easily applicable to most off- and on-policy DRL algorithms. We further propose an adaptive scaling method, which dynamically scales the effect of the distance measure for stabilizing the learning process and enhancing the overall performance.

## 2 DIVERSITY-DRIVEN STRATEGY

The main objective of the proposed diversity-driven exploration strategy is to encourage a DRL agent to explore different behaviors during the training phase. Diversity-driven exploration is an effective way to motivate an agent to examine a richer set of states, as well as provide it with an approach to escape from sub-optimal policies. This can be achieved by modifying the loss function $L_D$ as follows:

$$L_D = L - \mathbb{E}_{\pi' \in \Pi'}[\alpha D(\pi, \pi')] \tag{1}$$

where $L$ indicates the loss function of any arbitrary DRL algorithms, $\pi$ is the current policy, $\pi'$ is a policy sampled from a limited set of the most recent policies $\Pi'$, $D$ is a distance measure between $\pi$ and $\pi'$, and $\alpha$ is a scaling factor for $D$. The second term in eq. (1) encourages an agent to update $\pi$ with gradients towards directions such that $\pi$ diverges from the samples in $\Pi'$. Eq. (1) provides several favorable properties. First, it drives an agent to proactively attempt new policies, increasing the opportunities to visit novel states even in the absence of reward signals from $\mathcal{E}$. This property

is especially useful in sparse reward settings, where the reward is zero for most of the states in state space. Second, the distance measure $D$ motivates exploration by modifying an agent's current policy $\pi$, instead of altering its behavior randomly. Third, it allows an agent to perform either greedy or stochastic policies while exploring effectively in the training phase. For greedy policies, since $D$ requires an agent to adjust $\pi$ after each update, the greedy action for a state may change accordingly, potentially directing the agent to explore unseen states. This property also ensures that the agent acts consistently in the states it has been familiar with, as $\pi$ and $\pi'$ yield the same outcomes for those states. These three properties allow a DRL agent to explore an environment in a systematic and consistent manner.

## 2.1 IMPLEMENTATION OF OFF- AND ON- POLICY ALGORITHMS

We implement the proposed methodology on Deep Q-Network (DQN) Mnih (2015) and Advantage Actor-Critic (A2C), the synchronous version of Asynchronous Advantage Actor Critic Mnih (2016).

**DQN.** We make the following changes to the DQN algorithm. First, we additionally store the past Q-functions (denoted as $Q'(s, a)$) in replay buffer $Z$. Second, for the sake of defining a proper distance measure, we use a probabilistic formulation as in Plappert (2018) by applying the softmax function over the predicted Q-values. We therefore define $\pi(a|s) = \exp(Q(s, a))/\Sigma_i \exp(Q(s, i))$, where $i$ denotes the $i$-th action in state $s$. $\pi'(a|s)$ is defined similarly but uses $Q'(s, a)$ instead. We adopt KL-divergence as the distance measure, denoted as $D_{KL}$. Eq. (1) is rewritten as: $L_D = L - \mathbb{E}_{\hat{Q}(s,a) \sim U(Z)}[\alpha D_{KL}(\pi(a|s)||\pi'(a|s))]$, where $U(Z)$ represents the uniform samples from $Z$, $\alpha$ can be determined either by a predefined decay function, or by the adaptive method discussed in Section 2.2.

**A2C.** As A2C has no experience replay, we maintain the $n$ most recent policies for calculating the distance measure $D$. In general cases, $n = 5$ is sufficient to yield satisfactory performance. The loss function $L_D$ is thus expressed as: $L_D = L - \mathbb{E}_{s \sim \tau}[\alpha_i D_{KL}(\pi(s), \pi'_i(s))], i \in n$, where $\tau$ represents the batch of training data, and $\pi'_i$ is the $i$-th most recent policy. KL-divergence is used as the distance measure.

## 2.2 ADAPTIVE SCALING METHODS

Although $\alpha$ can be linearly annealed over time, we find this solution less than ideal in many practical settings. To update $\alpha$ in a way that leads to better overall performance, we consider two adaptive scaling methods: the distance-based and the performance-based methods. In our experiments, the off-policy algorithms use only the distance-based method, while the on-policy algorithm uses both methods for scaling $\alpha$.

**Distance-based.** Similar to Plappert (2018), we relate $\alpha$ to the distance measure $D$. We adaptively increase or decrease the value of $\alpha$ depending on whether $D$ is below or above a certain threshold $\delta$. The simple heuristic approach we use to update $\alpha$ for each training iteration is defined as:

$$\alpha := \begin{cases} 1.01\alpha, & \text{if } \mathbb{E}\big[D(\pi, \pi')\big] \leq \delta \\ 0.99\alpha, & \text{otherwise} \end{cases} \tag{2}$$

**Performance-based.** While the distance-based scaling method is straightforward and effective, it alone does not lead to the same performance for on-policy algorithms. The rationale behind this is that we only use the five most recent policies ($n = 5$) to compute $L_D$, which often results in high variance, and instability during the training phase. Off-policy algorithms do not suffer from this issue, as they can utilize experience replay to provide a sufficiently large set of past policies. Therefore, we propose to further adjust the value of $\alpha$ for on-policy algorithms according to the performance of past policies, in order to stabilize the learning process. We define $\alpha_i$ in one of the following two strategies:

$$\alpha_i := -(2(\frac{P(\pi'_i) - P_{min}}{P_{max} - P_{min}}) - 1) \text{ (Proactive)}; \quad \alpha_i := 1.0 - \frac{P(\pi'_i) - P_{min}}{P_{max} - P_{min}} \text{ (Reactive)} \tag{3}$$

where $i \in n$, $P(\pi'_i)$ denotes the average score of $\pi'_i$ over five episodes, and $P_{min}$ and $P_{max}$ represent the minimum and maximum scores attained by the set of past policies $\Pi'$. Note that $\alpha_i$ falls in the interval $[-1, 1]$ for the proactive strategy, and $[0, 1]$ for the reactive one. The proactive strategy incentivizes the current policy $\pi$ to converge to the high-performing policies in $\Pi'$, while keeping away from the poor ones. On the other hand, the reactive strategy only motivates $\pi$ to stay away from the underperforming policies. Please note that we apply both eq. (2) and eq. (3) to the on-policy methods in our experiments.

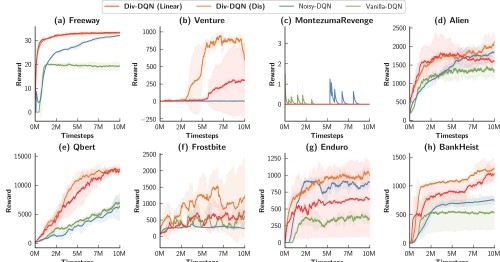 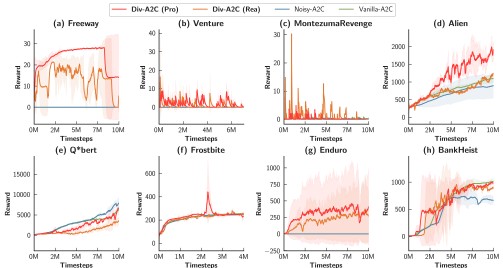

Figure 1: Learning curves comparison of different DQNs in Atari 2600.

Figure 2: Learning curves comparison of different A2Cs in Atari 2600.

# 3 EXPERIMENTAL RESULTS ON ATARI 2600

We evaluate our method against the other baselines on Atari 2600. The baseline models we used for comparison include DQN, A2C, and their Noisy Net variants Fortunato (2018). All of the implementations are developed based on OpenAI baselines[1]. We provide in-depth discussions about the empirical results of a few selected games from both the hard and easy exploration categories, according to the taxonomy presented in Bellemare (2016). Each agent is trained with 40M frames, and the overall performance is estimated over three random seeds. In Figs. 1 and 2, we plot the median scores during the training phase. Div-DQN/A2C stand for diversity-driven DQN/A2C. Div-DQN-(Linear/Dis) denote Div-DQN with linear decay/distance-based scaling, respectively. Div-A2C-(Pro/Rea) stand for Div-A2C with proactive/reactive scaling. Noisy-DQN/A2C represent NoisyNet DQN/A2C. Vanilla DQN/A2C denote the original versions of DQN/A2C.

To validate the effectiveness of our method on sparse and deceptive reward settings, we select a few games from the hard exploration category for evaluation. Fig. 1 plots the learning curves of all the models considered in the training phase. It can be seen that our method demonstrates superior or comparable performance to the baseline models in all games. Particularly, we observed that our diversity-driven strategy helps an agent explore a wider area more efficiently compared to the other baselines. This property is especially useful when the state spaces become sufficiently large. For example, in *Freeway*, we noticed that our agents are able to quickly discover the only reward on the other side of the road, while the other baselines always fail at the starting point. This observation is consistent with the learning curves illustrated in Figs. 1 and 2, where Div-DQN and Div-A2C learn considerably faster and better than the baseline models. In *Q*bert*, we found that our Div-DQN agent utilizes the discs, which transport the player to the top of the pyramid, as an escape from danger, whereas the other models never learn to avoid enemies using those discs. In addition to the efficiency, we also found that our method is more systematic compared to the others. Our agents are frequently able to discover new states that have yet been visited. As shown in Fig. 1, Div-DQN is the only one that learns a successful policy in *Venture*. On the contrary, DQN, A2C, Div-A2C, Noisy-DQN, and Noisy-A2C agents are often hit by the monsters and lost the game frequently, as they employ random exploration strategies. We also found that our method effectively motivates the agent to bypass small (deceptive) rewards, and look for alternate ways to reach the optimal goal. For instance, in *Alien*, our agents learn to collect rewards and detour from the aliens, while the baselines only focus on the immediate rewards in front of them without taking the aliens into consideration. This enables our agents to obtain higher average rewards than the others, as shown in Figs. 1 and 2. In *Frostbite*, both Div-DQN and Div-A2C agents quickly discover the optimal strategy. The agents first jump on the ice plates to build an igloo, then approach the shore, and enter the igloo to obtain the highest reward. The other baselines, on the other hand, tend to collect small rewards by jumping around the ice plates. These results indicate that our method is effective in preventing a policy from converging prematurely to local optima. To sum up, our method results in an improvement in terms of scores and training efficiency for the hard exploration games except for *Montezuma's Revenge*, which is well known for its complexity and difficulty. We also show that the proposed diversity-driven exploration strategy can improve the training efficiency for games in the easy exploration category as well. From the learning curve of *Enduro* presented in Figs. 1 and 2, it can be seen that our Div-DQN and Div-A2C agents learn significantly faster than the baseline models, showing a considerable improvement in the overall performance.

---

[1]https://github.com/openai/baselines

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
