# OpenReview forum: "Diversity-Driven Exploration Strategy for Deep Reinforcement Learning"
_ICLR.cc/2018/Workshop — Accept_

### Official Review · AnonReviewer3 · 2018-03-06

**Rating:** 7
**Confidence:** 4

**Review:**

# Summary
This paper proposes a new method for exploration in reinforcement learning. The idea is to encourage the policy to be different from its recent policies by introducing KL-divergence penalty between the current policy and the past policies. The result shows that the proposed method (applied to DQN and A2C) performs better than baselines on several Atari games.

[Pros]
- Interesting idea for improving exploration in RL.
- A promising result on Atari games.

[Cons]
- No theoretical result.

# Novelty
- The proposed idea is novel to my knowledge.

# Technical correctness
- It might be good to have some theory that justifies the proposed approach (if possible). Though the idea looks reasonable, the policy with the proposed diversity-driven objective might oscillate without deeply exploring new policies by making a cycle of policies that are reasonably different from each other.

# Quality
- The result looks promising on some of the Atari games (e.g., Enduro). A more extensive result on many Atari games and other domains would further strengthen the paper.

---

### Official Review · AnonReviewer2 · 2018-03-09
**Interesting method, insufficient experimental validation, limited related work**

**Rating:** 6
**Confidence:** 3

**Review:**

This paper proposes a diversity penalty which penalises policies for being similar to previous policies.
The diversity penalty is formulated as a KL divergence between the previous policies and the current policy, thus encouraging the agent to explore new policies.

This seems like a reasonable objective for exploration in policy space.

There are two main concerns:
1) Novelty:
The paper misses proper comparisons to other exploration methods eg [1], [2]. This raises the question whether other, more closely related publications have been missed which could substantially threaten algorithmic novelty.
2) Experimental validation:
The Atari experiments are far from conclusive. The authors test on only 3 random seeds and lack statistical validation of the results. It would also be nice to have other exploration focussed baselines included in the validation.

[1]: Reinforcement Learning with Deep Energy-Based Policies (Haarnoja T., Tang H., Abbeel P., Levine S. ICML 2017)
[2]: Equivalence Between Policy Gradients and Soft Q-Learning ( Schulman, J., Abbeel, P. and Chen, X., arXiv preprint arXiv:1704.06440, 2017).

---

### Official Review · AnonReviewer1 · 2018-03-10
**An interesting exploration heuristic with some good Atari results**

**Rating:** 6
**Confidence:** 4

**Review:**

This paper presents a diversity-driven exploration strategy for reinforcement learning.
This algorithm provides a bonus whenever the policy of the algorithm differs from the "baseline" policy that has been recently observed.

There are several things to like about this paper:
- The approach outlines a reasonable exploration heuristic for RL.
- The paper outlines practical+code implementation details that may inspire people to build on this.
- The results on Atari seem to outperform state of the art baselines.

On the other hand the paper has several shortcomings:
- The algorithm is not clearly presented as a *heuristic* and a reader might easily be confused into thinking that this algorithm doesn't have clear shortcomings.
- Discussion or understanding of how/why this relates to literature on "efficient exploration" or why they are not relevant could be improved.
- Some of the discussion of individual Atari game behavior is confusing without video/illustration.

Overall I think that this paper is reasonable, and would probably be interesting to the conference.

---

### Decision · Program_Chairs · 2018-03-20
**ICLR 2018 Workshop Acceptance Decision**

**Decision:**

Accept

**Comment:**

Congratulations, your paper was accepted to the ICLR workshop.